# Addressing the Reciprocal Crosstalk between the AR and the PI3K/AKT/mTOR Signaling Pathways for Prostate Cancer Treatment

**DOI:** 10.3390/ijms24032289

**Published:** 2023-01-24

**Authors:** Fabio Raith, Daniel H. O’Donovan, Clara Lemos, Oliver Politz, Bernard Haendler

**Affiliations:** 1Research & Development, Pharmaceuticals, Bayer AG, Müllerstr. 178, 13353 Berlin, Germany; 2Bayer Research and Innovation Center, Bayer US LLC, 238 Main Street, Cambridge, MA 02142, USA

**Keywords:** prostate cancer, androgen receptor, PI3K, AKT, mTOR

## Abstract

The reduction in androgen synthesis and the blockade of the androgen receptor (AR) function by chemical castration and AR signaling inhibitors represent the main treatment lines for the initial stages of prostate cancer. Unfortunately, resistance mechanisms ultimately develop due to alterations in the AR pathway, such as gene amplification or mutations, and also the emergence of alternative pathways that render the tumor less or, more rarely, completely independent of androgen activation. An essential oncogenic axis activated in prostate cancer is the phosphatidylinositol-3-kinase (PI3K)/AKT/mammalian target of rapamycin (mTOR) pathway, as evidenced by the frequent alterations of the negative regulator phosphatase and tensin homolog (PTEN) and by the activating mutations in PI3K subunits. Additionally, crosstalk and reciprocal feedback loops between androgen signaling and the PI3K/AKT/mTOR signaling cascade that activate pro-survival signals and play an essential role in disease recurrence and progression have been evidenced. Inhibitors addressing different players of the PI3K/AKT/mTOR pathway have been evaluated in the clinic. Only a limited benefit has been reported in prostate cancer up to now due to the associated side effects, so novel combination approaches and biomarkers predictive of patient response are urgently needed. Here, we reviewed recent data on the crosstalk between AR signaling and the PI3K/AKT/mTOR pathway, the selective inhibitors identified, and the most advanced clinical studies, with a focus on combination treatments. A deeper understanding of the complex molecular mechanisms involved in disease progression and treatment resistance is essential to further guide therapeutic approaches with improved outcomes.

## 1. Introduction

Prostate cancer is the second most frequently diagnosed tumor and the sixth leading cause of cancer death in men, with 1,414,259 new cases and 375,304 deaths reported worldwide in 2020 [1]. The incidence rates and mortality have remained stable for several years, following a sharp decline linked to prostate-specific antigen testing [2]. The main risk factors are age and ethnicity, with family history and diet playing additional important roles [3,4]. Germline mutations are altogether rare in prostate cancer and affect mainly the genes involved in DNA damage response [5,6].

Men with localized or regionally spread prostate cancer have a favorable prognosis and may first undergo prostatectomy or radiotherapy [7]. Once the disease progresses, hormone-sensitive prostate cancer (HSPC) is treated by orchiectomy or chemical castration using gonadotropin-releasing hormone (GnRH) agonists or antagonists to suppress androgen synthesis in the testes [6,8]. A complete androgen blockade is achieved with the cytochrome P450 17A1 (CYP17A) inhibitor abiraterone acetate, which locally reduces androgen synthesis, and with competitive antagonists of androgen receptor (AR) function [6,8]. Despite an initial good response, therapy resistance leading to castration-resistant prostate cancer (CRPC) ultimately occurs, for which the outcome is poor. Resistance mechanisms include *AR* gene amplification and overexpression, the emergence of AR mutations and splice variants, as well as increased intra-tumoral androgen synthesis [6,8]. Late-stage treatment options include taxanes, radium-223, poly (ADP-ribose) polymerase (PARP) inhibitors, autologous cellular immunotherapy with sipuleucel-T, and more recently, the radioconjugate ^177^Lu-PSMA-617 [9]. The anti-PD-1 antibody pembrolizumab, which received tumor-agnostic approval for patients with high microsatellite instability, is used for the treatment of metastatic CRPC (mCRPC) patients [9], but immune checkpoint inhibitors have so far shown little benefit as single agents or in combination studies in unselected prostate cancer patients [6]. Only a few direct comparisons between all of these treatments exist, making individual therapy choices difficult so predictive biomarkers are highly needed [9]. The transformation to neuroendocrine prostate cancer (NEPC), a lethal form of prostate cancer, represents an increasing form of therapeutic resistance characterized by little or no AR activity and high levels of neuroendocrine markers [10,11,12]. Unfortunately, no effective treatment options are currently available for NEPC, but the recent identification of different subtypes based on genomic and transcriptomic classification has the potential to guide future therapy decisions [13,14,15].

The genomic profiling of samples from patients at different stages of prostate cancer led to the identification of numerous alterations with varying frequencies along disease progression [6,16]. The main somatic change reported in localized prostate cancer is *Ets-related gene* (*ERG*) fusion with androgen-responsive promoters, which is detected in nearly 50% of cases in Western patients but far less frequently in men of Asian or African descent [6,17]. Next, come *phosphatase and tensin homolog* (*PTEN*) deletions, inactivating mutations, and silencing by promoter methylation, with increased frequencies along disease progression [6,18,19,20]. In addition, activating mutations of the phosphatidylinositol-3-kinase (PI3K) can also be observed, mainly in mCRPC, so that the oncogenic activation of the PI3K/AKT/mammalian target of rapamycin (mTOR) pathway represents an important event involved in resistance to androgen deprivation therapy (ADT) [19,21]. As AR overexpression and mutations are also detected in the majority of mCRPC patients [8,16], increasing attention has been given to the molecular mechanisms underlying the reciprocal crosstalk and feedback mechanisms between these two pathways and their role in tumor progression. Recently, a novel prostate cancer classification based on gene expression profiles from the PI3K and AR pathways has been proposed, and a worse progression-free survival was evidenced for the subgroup with a mixed PI3K and AR signature [22].

## 2. AR Signaling in Prostate Cancer

### 2.1. AR Pathway Overview

Androgens play an essential role in spermatogenesis and in the development of male secondary sexual characteristics [23]. Androgen synthesis is under the control of the hypothalamic-pituitary-gonadal axis, starting with the gonadotropin-releasing hormone, which stimulates the synthesis and secretion of the follicle-stimulating hormone and the luteinizing hormone [24,25]. This promotes spermatogenesis and androgen production in the testis. Testosterone is the primary male sex hormone and is converted by 5-alpha-reductase to the more potent dihydrotestosterone in the prostate and in other tissues [26].

Androgens act by binding to the cytoplasmic AR, which is a member of the steroid hormone receptor family. This is followed by the shedding of heat-shock proteins and chaperones, conformational changes, nuclear translocation, homodimerization, and binding to specific DNA response elements. The AR dimer forms complexes with a range of pioneer factors and coactivator proteins [4,27,28,29]. This activation chain ultimately induces a dramatic change in downstream gene expression and corresponding protein levels, which can be reverted by treatment with an AR pathway inhibitor [30,31,32,33].

Three functional domains and a hinge region have been identified in the AR [34,35,36]. The ligand-binding domain is bound by androgens and also by the approved competitive antagonists and contains the transactivation region AF-2 [8,37,38]. The DNA-binding domain is highly conserved among steroid receptors and is responsible for the recognition and binding to consensus DNA motifs named androgen-responsive elements (AREs) via two zinc finger regions. It also contains the dimerization surface D-box. The N-terminal domain harbors the important transactivation regions TAU-1 and TAU-5 and short motifs involved in coactivator recruitment. Inter- and intramolecular interactions mediated by specific motifs located in different AR domains play important functional roles [39]. The AR also undergoes post-translational modifications, such as phosphorylation or acetylation, at specific sites, but the overall impact on normal physiology and pathology is not clearly understood [40,41,42,43].

Recent studies indicate that the AR N-terminal domain contains flexible, intrinsically disordered regions, which are also found in other transcription factors and involved in the formation of liquid condensates [44,45]. The N-terminal domain has a predominant role in AR phase separation in the nucleus, but other regions are involved as well [46]. This mechanism allows high local AR concentration followed by sustained downstream gene expression [35,44,45,47] and may also impact the efficacy of inhibitors [46]. Conformational changes in the three-dimensional chromatin structure and the emergence of topologically associating domains favoring promoter and enhancer interactions have furthermore been described as essential components involved in AR target gene regulation [35,48,49,50].

### 2.2. AR Reprogramming in Prostate Cancer

Adenocarcinoma is the most frequently diagnosed form of prostate cancer, and the essential role of AR signaling is evidenced by the different approved therapies addressing this pathway (see above). Understanding the differences between normal AR signaling as needed for the proper physiological function and abnormal signaling eliciting pathogenesis has been the subject of intensive research efforts in recent years. The reprogramming of the AR cistrome leading to increased plasticity, de-differentiation, and the reactivation of developmental programs is associated with tumor progression and poor outcomes [51,52,53]. Differences in AR cistromes have been reported between early and late-stage prostate cancer samples [54,55,56]. AR-V7 is a constitutively active splice variant lacking the ligand-binding domain and found predominantly in CRPC [57]. Its cistrome differs from that of full-length AR and also undergoes changes along prostate cancer progression [58,59]. The cistromes of the forkhead box protein A1 (FOXA1) and homeobox protein HOXB13, two essential pioneer factors associated with the AR and AR-V7, are also altered in prostate tumors [52,58]. Interestingly, AR-binding sites are highly mutated in clinical samples of prostate cancer, and this may affect enhancer activity [60]. Therapy resistance has been linked to the reprogramming of the AR cistrome and involves epigenetic regulators [61]. AR overexpression, a frequent resistance mechanism in CRPC, leads to modifications in DNA binding [62]. Changes in the cistrome of AR splice variants also take place, and preferential binding sites impacting downstream regulated genes have been described [63].

The comparison of AR transcriptomes between normal prostate and samples from different tumor stages indicates dramatic differences. Gene sets corresponding to AR-binding sites lost or gained between normal prostate and tumors have been defined, and their association with treatment outcomes analyzed [53]. In another study, different subgroups based on AR activity levels have been defined [64]. AR target genes selectively regulated between untreated and treated prostate tumors or associated with response or resistance to therapy have been reported [54,65]. RNA sequencing performed on samples from CRPC patients treated with enzalutamide revealed that non-responders expressed gene sets linked to low AR activity and to a stemness program [66]. AR-V7 preferentially associates with corepressors, and transcriptomic data show that it is a negative regulator of antiproliferative genes [67].

### 2.3. Neuroendocrine Prostate Cancer

De novo neuroendocrine prostate cancer is rarely observed, but treatment-emergent forms are now more frequently diagnosed due to the increased and earlier use of potent AR signaling inhibitors [10,11,12]. Numerous genetic aberrations, as well as expression changes in transcription factors and epigenetic regulators, are involved in NEPC emergence, and AR expression is low or undetectable [10,11,12,68,69]. Neuroendocrine-specific DNA regulatory elements recognized by the FOXA1 pioneer factor following local changes in histone modifications have been evidenced [68]. RNA sequencing and the histological assessment of neuroendocrine prostate cancer have allowed detailed phenotypic characterization. Neuroendocrine tumors are mainly of the basal subtype, and a subgroup leading to liver metastases has furthermore been described [14,15]. A chromatin accessibility approach combined with transcriptomic analysis confirmed the existence of a neuroendocrine subgroup beside three other CRPC subgroups [13].

## 3. PI3K/AKT/mTOR Signaling in Prostate Cancer

### 3.1. PI3K/AKT/mTOR Pathway Overview

PI3K/AKT/mTOR signaling plays an essential role in prostate cancer and in resistance to therapies. Genetic alterations are common and mainly affect the *PTEN* gene, which is the major regulator of the pathway [70,71,72]. Moreover, numerous mutations in the catalytic and regulatory subunits of PI3K complexes and in the downstream effectors have been evidenced [21,71,72].

The PI3K family of lipid kinases is at the interface between extracellular activation signals and intracellular pathways controlling multiple processes. Different classes have been defined based on the structure and function of the individual family members. Class IA and IB members are the best studied regarding their role in prostate cancer [19,21]. They form heterodimers composed of a catalytic p110 subunit and a smaller regulatory subunit. Class IA includes three catalytic subunits, named p110α, p110β, and p110δ, and five regulatory subunits named p85α, p55α, p50α, p85β, and p55γ, while class IB is composed of a single catalytic subunit p110γ, that binds to the p101-p84 regulatory subunits. Class I PI3 kinases are mainly activated by extracellular signals that trigger growth factor receptors and by the Ras GTPase, leading to their recruitment to the plasma membrane where they phosphorylate phosphatidylinositol-4,5-biphosphate (PI(4,5)P_2_) to produce phosphatidylinositol-3,4,5-triphosphate (PI(3,4,5)P_3_). This is followed by the downstream activation of several kinases, including 3-phosphoinositide-dependent protein kinase-1 (PDK1) and the mTOR complex 2 (mTORC2), which then sequentially phosphorylates the serine/threonine kinase AKT at two different sites [73]. There are three related AKT isoforms with a broad tissue distribution and largely overlapping functions, but non-redundant roles have been defined in different tumor types, including prostate cancer [74]. Activated AKT phosphorylates multiple downstream proteins at a specific consensus motif, thus leading to stimulation or inhibition [75]. This, in turn, prompts the regulation of many cellular processes, including proliferation, invasion, survival, and apoptosis [21]. The protein complexes mTOR complex 1 (mTORC1) and mTORC2 play important, distinct tasks in the AKT signaling pathway. mTORC1 stimulates transcription and translation and contains several proteins, including the regulatory-associated protein of mTOR (RAPTOR) and the DEP domain-containing mTOR-interacting protein (DEPTOR), which is often overexpressed in late-stage prostate cancer. mTORC2 is involved in multiple aspects of cell proliferation and survival and is composed of the rapamycin-insensitive companion of mTOR (RICTOR), DEPTOR, and a few other proteins [19,76]. It promotes the autophosphorylation of multiple kinases, including AKT, for their full activation [77]. The forkhead box O (FOXO) transcription factors are further important regulatory components of the PI3K/AKT/mTOR pathway, and they are inactivated following phosphorylation by AKT [19].

Members of the more distantly related phosphatidylinositol phosphate kinase (PIPK) superfamily, which phosphorylate phosphoinositides play a role in prostate cancer as well [78]. PIKfyve is responsible for the generation of phosphatidylinositol 3,5-biphosphate (PI(3,5)P_2_), and its blockade or expression knock-down in different prostate cancer models leads to reduced autophagy and potentiates the response to immune checkpoint inhibitors [79]. The lipid kinase phosphatidylinositol 4-phosphate 5 kinase PIP5K1α is involved in prostate cancer cell growth as evidenced by expression knock-down and inhibition [80], and this was shown to involve its N-terminal domain by using in vivo xenograft studies [81].

PTEN is a critical regulator of the PI3K/AKT/mTOR signaling cascade, which acts as a gatekeeper by dephosphorylating PI(3,4,5)P_3_ to PI(4,5)P_2_ [70]. *PTEN* gene inactivation is observed in many tumor types, including prostate cancer, and can occur by diverse mechanisms, ultimately leading to the constitutive activity of the PI3K/AKT/mTOR pathway. Inositol polyphosphate 4-phosphatase type II (INPP4B) is another important regulator of the PI3K/AKT/mTOR pathway, which is responsible for the dephosphorylation of PI(3,4)P_2_ [82].

In addition to its pro-survival and anti-apoptotic impact, the PI3K/AKT/mTOR pathway is also an important player in cellular metabolism [21]. The role of mitochondrial rewiring in prostate cancer was shown by the transcriptomic and proteomic analysis of patient-derived models [83]. Metabolic rewiring is also seen after the activation of mTOR, which is followed by important changes in anabolic processes [84]. This includes increased fatty acid synthesis and uptake, which are both essential to the fueling of prostate cancer cells which are particularly dependent on lipid metabolism for growth, especially at advanced stages [85,86,87].

### 3.2. PI3K Alterations in Prostate Cancer

PI3K mutations are observed both in the context of the *PTEN* gene wild-type and *PTEN* gene loss. An analysis of a large number of prostate cancer models shows frequent alterations of p110α and p110β and feedback between the isoforms [88]. Hotspots for activating mutations that are responsible for unregulated cell proliferation have been revealed in different regions of p110α, including the helical and kinase domains [89,90]. A recent study reported that elevation in specific circulating lipids measured in mCRPC patients combined with different genetic abnormalities, including PI3K alterations, are associated with decreased overall survival [91]. Mutations in p110δ are less frequent and mainly found in mCRPC [19,92]. Importantly, PI3K mutations act as oncogenic drivers in prostate cancer and cooperate with *PTEN* loss to accelerate disease progression [93]. Driver mutations in catalytic subunits reduce autoinhibition and thereby cause the exposure of the kinase domain or an increase in membrane interaction, ultimately leading to enhanced activity [94,95].

Reduced expression and inactivating mutations have furthermore been reported in prostate cancer for the repressive PIK3R1 regulatory unit, mainly at the metastatic stage, and are associated with unfavorable clinical outcomes [88,96].

### 3.3. AKT Alterations in Prostate Cancer

*AKT* gene amplification is observed in 2–5% of advanced prostate cancer cases, but activating mutations are rare [19]. The analysis of cell-free DNA from mCRPC patients reveals a 6% incidence of activating mutations in *AKT* or *PIK3CA* (which encodes the p110α catalytic subunit) and a negative correlation with *AR* gene copy gain [97]. Additionally, *AKT1* activating mutations are mutually exclusive with *PTEN* alterations [97]. Importantly, the AKT regulator PDK1 is stabilized by the E3 ligase speckle-type POZ protein (SPOP), which is mutated in about 15% of early and late-stage prostate cancer [98]. Mouse bearing the SPOP mutation F133V which is the most frequently found in prostate cancer patients, show little changes in their prostate [99]. The situation is, however, different in a *PTEN* loss background where invasive prostate cancer develops [99].

The role of AKT in NEPC has furthermore been reported. Early studies in prostate cancer cell lines showed AKT activation to promote neuroendocrine differentiation [100,101]. Another work describes that constitutively activated AKT, together with c-Myc, induces the de novo neuroendocrine differentiation of luminal epithelial cells from the prostate [102]. The AKT3 isoform was highlighted as playing an essential role in neuroendocrine differentiation based on overexpression and co-localization with marker genes [103]. A survey of available genomic and RNA-seq data of human NEPC samples revealed frequent alterations in the PI3K/AKT/mTOR pathway, mainly in the *PTEN* gene, but no significant differences in the gene expression levels of *AKT* isoforms were observed [104].

### 3.4. mTOR Complex Alterations in Prostate Cancer

mTOR is a central hub of metabolic sensors with a key role in cell survival and proliferation. A proteomic profiling of the mTOR complex bound to chromatin in different prostate cancer cell lines shows the androgen-dependent binding of the AR and the additional important function of the nucleosome remodeling deacetylase NuRD complex for the regulation of gene expression [105]. Different components of mTOR-associated complexes are modified in prostate cancer but usually in small frequencies. The most frequently altered mTOR complex component is DEPTOR which is amplified in about 5% of mCRPC cases, and this is linked to worse disease outcomes [19]. DEPTOR suppresses the activity of both mTORC1 and mTORC2 [106] but activates AKT via negative feedback [107,108]. On the other hand, a tumor suppressor action has also been described, and a mouse *DEPTOR* knock-out model was recently reported to develop prostate tumorigenesis, suggesting a dual, context-dependent action [109]. RICTOR is involved in prostate cancer progression in the *PTEN*-deficient context via the stimulation of AKT phosphorylation [110,111,112]. As mentioned, prostate cancer growth is particularly dependent on altered lipid metabolism, a process controlled at least in part by mTOR.

### 3.5. FOXO Alterations in Prostate Cancer

The FOXO transcription factors are important downstream targets of AKT. Gene deletion of the tumor suppressors *FOXO1* and *FOXO3* is frequently observed in late-stage prostate cancer [19]. FOXO1 has an inhibitory impact on prostate cancer which is lost following phosphorylation and cytosolic migration [113]. The binding of FOXO1 to the AR N-terminal domain, followed by the impaired recruitment of co-activators and loss of activity, has been evidenced [114]. FOXO1 also binds to and inhibits the ERG transcription factor, which is overexpressed in about 50% of early prostate tumors, and its deletion promotes tumors in mice with high ERG levels [115]. FOXO activation after the knock-out of *AKT1* and *AKT2* impairs AR nuclear translocation [116]. The modulation of FOXO3 activity and levels affects the growth and proliferation of numerous prostate cancer cell lines [117]. In vivo, the repression of FOXO3 activity leads to increased prostate cancer progression in the transgenic adenocarcinoma of the mouse prostate (TRAMP) mice [118].

### 3.6. PTEN Loss in Prostate Cancer

*PTEN* inactivation is the main genetic alteration reported in prostate cancer. Genomic homozygous deletions and somatic inactivating alterations, such as truncations or frameshift mutations, are observed in about 20–30% of primary tumors and 40–60% of mCRPC cases [70]. Regulatory action of some microRNAs in controlling *PTEN* gene expression and protein ubiquitylation leads to diminished activity and has been reported [119,120]. Both homozygous and heterozygous *PTEN* loss is associated with biochemical recurrence, quicker tumor relapse, and reduced responses to therapy at different stages of prostate cancer [70]. Importantly, *PTEN* loss is often accompanied by inactivating mutations of other tumor suppressor genes, such as those for *retinoblastoma* (*Rb*) or for *TP53*, or by the activation of oncogenes, altogether inducing stronger tumor progression, metastasis formation, and resistance to ADT [121]. *PTEN*-deficient prostate cancer patients often also harbor inactivating mutations in *lysine N-methyltransferase 2C* (*KMT2C*) and mice bearing both alterations develop prostate cancer metastases with a hallmark *c-Myc* gene signature [122]. The expression loss of *PTEN* and other tumor suppressors based on 5-hydroxymethylcytosine counts over the corresponding gene body is part of a signature that is predictive of worse survival in mCRPC patients [123].

The effect of *PTEN* loss on tumor growth and survival has been evidenced in numerous experimental prostate cancer models [124]. Mice with conditional, selective knock-out of the *PTEN* gene in the prostate develop invasive tumors that recapitulate many aspects observed in patients [125,126]. *PTEN* haploinsufficiency leads to prostate lesions in mouse transgenic models but not to malignant tumors, so this alteration is often combined with additional genetic changes such as *Rb* or *p53* inactivation in tumor models to mimic more closely the clinical situation [124]. Primary tumors and epithelial cell lines derived from mouse prostate cancer with *PTEN* deletion respond similarly to PI3K inhibition [127]. Several studies with different preclinical models, including genetically engineered mice, suggest that p110β is the main driver in *PTEN*-deficient models of prostate cancer [128,129,130]. However, a recent detailed analysis of patient-derived prostate cancer organoids reported that *PTEN* loss did not cause selective sensitivity to PI3K isoform inhibitors and that changes in isoform dependency might take place [88]. Additionally, AKT inhibition is more effective than PI3K blockade in the context of *PTEN* loss [88].

The first attempts to restore function by the delivery of recombinant PTEN protein or mRNA, by targeting negative regulators or microRNAs, by transcriptional reactivation using CRISPR/Cas9 technology, or by reverting the epigenetic silencing of the corresponding promoter have shown positive results in preclinical models [70].

## 4. Crosstalk between PI3K/AKT/mTOR and AR Signaling

Reciprocal feedback between the PI3K/AKT/mTOR and AR pathways has been evidenced in multiple preclinical models, with the blockade of one pathway leading to the stimulation of the other one (Figure 1) [21,131,132,133]. This has led to the extensive evaluation of dual inhibitory approaches initially in preclinical models and, more recently, in patients with late-stage prostate cancer [21,133,134,135].

### 4.1. AR Signaling Impacts the PI3K/AK/mTOR Pathway

There is a negative regulatory effect of AR signaling on AKT function, and AR overexpression reduces AKT activity [136]. The castration of mice with conditional *PTEN* deletion leads to an elevation in the expression and phosphorylation of AKT in prostate tumor cells [136]. Several other studies also show that androgen ablation elicits enhanced AKT activity, which, in turn, impaired the anti-tumor efficacy of AR inhibitors in preclinical models [137,138,139,140]. The role of the PH domain and leucine-rich repeat protein phosphatase (PHLPP)/FK506-binding protein 51 (FKBP51)/I kappa B kinase α (IKKα) complex in the crosstalk between AR and AKT signaling was evidenced [140,141]. The downregulation of *FKBP5* gene expression by *AR* deletion prevented the inhibitory effects of PHLPP on AKT [140]. Very recently, the isomerase activity of the corresponding protein FKBP51 was additionally shown to be essential for AR dimerization and function [142].

Androgens stimulate mTOR activity in prostate cancer cells with *PTEN* deficiency by the upregulation of genes involved in nutrient availability and independently of PI3K/AKT activation [143]. AR signaling also impacts the cellular metabolism reprogramming observed in prostate cancer via binding, together with mTOR, to regulatory regions of the *sterol regulatory element-binding transcription factor 1* (*SREBF1*) gene, which leads to increased expression [144]. Androgen stimulation additionally enhances SREBF1 stimulation of its target genes, including the essential lipogenic player *fatty acid synthase* (*FASN*), and this is prevented by mTOR inhibitors [144]. Activated AR also directly stimulates the expression of *FASN* and of numerous other genes involved in the fatty acid synthesis, and this promotes tumor growth, as evidenced in a variety of preclinical models [86]. The androgen stimulation of metabolic reprogramming is not seen after the inhibition of mTOR activity [145]. Following androgen deprivation of an orthotopic mouse *PTEN*-deleted prostate tumor, the stimulation of AKT led to elevated levels of mitochondrial hexokinase 2 (HK2), and metabolic reprogramming was observed [146]. The impact of AR blockade on the lipid profiles of prostate cancer patients has furthermore been outlined [147].

### 4.2. The PI3K/AKT/mTOR Pathway Impacts AR Signaling

Most data on the importance of PI3K and AKT in the modulation of AR function and effects on prostate cancer proliferation were generated with specific inhibitors, thus providing strong evidence for the relevance of the pathway for therapeutic targeting [148]. Several studies using PI3K and AKT inhibitors revealed a remarkable impact on prostate cancer cell lines and patient-derived xenografts, also in castration-resistant models [130,131,149,150,151,152]. The dual inhibition of PI3K and mTOR led to an enhanced AR target gene expression in a *PTEN*-deficient cell line and stabilization of the AR by HER kinases [138]. The blockade of a CRPC cell line by an AKT inhibitor was found to be transient and linked to the increased AR binding and activation of AR target genes, and this was overcome by additional treatment with an anti-androgen [153]. A compensatory rise of glucocorticoid receptor expression following AKT inhibitor treatment was identified by another group, and this was also prevented by AR inhibition [151]. Indeed, several reports support the superior efficacy of combining inhibitors of the PI3K/AKT/mTOR and AR signaling pathways in reducing tumor growth, both in vitro and in vivo preclinical studies [124,131,150,154].

Concerning mTOR, direct interaction with the AR following nuclear translocation eliciting transcriptional reprogramming has been reported [76,105]. Importantly, an mTOR gene signature that discriminates between normal and tumor prostate and that is predictive of progression has been proposed [76].

Another approach to assess crosstalk is the downregulation or overexpression of regulators of the PI3K/AKT/mTOR pathway. The deletion of *PTEN* in prostate cancer cells is followed by diminished androgen-dependent gene expression and progression to androgen-independent proliferation [140]. A direct effect on endogenous AR expression is furthermore observed in *PTEN*-deleted mice [136]. The atypical mitogen-activated protein kinase MAPK4 activates the AKT/mTOR signaling pathway and transforms prostate epithelial cells [155]. MAPK4 also enhances AR function via the stimulation of *GATA2* gene expression and protein stability, thus leading to increased prostate cancer cell proliferation [156]. In line with this, MAPK4 overexpression stimulates, whereas its knock-down inhibits the growth of prostate cancer xenografts in vivo [156]. The histone deacetylase HDAC3 is another common activator of the AKT/mTOR and AR pathways. It increases AKT phosphorylation, whereas its conditional deletion or inhibition in *PTEN* knock-out mice reduces both AKT and AR signaling [157]. HDAC3 knockdown or blockade causes the impaired growth of prostate cancer organoids and xenografts in the context of *PTEN* deletion or *SPOP* mutation [157]. Reduced INPP4B levels are seen in prostate cancer, and overexpression leads to the inactivation of the PI3K/AKT/mTOR pathway and suppression of prostate cancer cell migration and invasion [158]. This was confirmed in another study which documented how INPP4B impacts the AR transcriptional program and that its depletion stimulates prostate cancer cell proliferation [159]. The zinc finger homeobox protein 3 (ZFHX3) is a transcription factor often mutated in prostate cancer and it suppresses PI3K/AKT/mTOR signaling [160]. Its expression is under androgen control [161], and the concomitant deletion of *ZFHX3* and *PTEN* leads to increased prostate neoplasia in a mouse model [162]. 5′-adenosine monophosphate-activated protein kinase (AMPK) is a target phosphorylated by AKT and an upstream regulator of mTOR involved in prostate cancer metabolism, but its impact on cell survival remains complex due to the numerous pathways it interacts with [163]. A highly specific AMPK inhibitor has recently been shown to have strong anti-proliferative action on androgen-dependent prostate cancer cell lines and it down-regulates the expression of several genes involved in lipid metabolism [164].

## 5. AR Inhibitors

### 5.1. Approved and Clinically Advanced AR Inhibitors

The first generation of AR inhibitors approved by the American Food and Drug Administration (FDA) for prostate cancer treatment included bicalutamide, flutamide, and nilutamide, which were launched between 1989 and 1996 [38]. However, the response to these compounds is of limited duration, so more potent second-generation AR inhibitors have later been developed. In recent years, the FDA has approved three new competitive AR antagonists for the treatment of prostate cancer [165]. The FDA granted approval of enzalutamide for the treatment of mCRPC in 2012, for non-metastatic castration-sensitive prostate cancer in 2018, and for metastatic castration-sensitive prostate cancer in 2019 [166]. Apalutamide was approved for non-metastatic CRPC in 2018 and for metastatic castration-sensitive prostate cancer in 2019 [167]. Darolutamide achieved FDA approval for the treatment of non-metastatic CRPC in 2019 [168,169] and very recently for metastatic HSPC, in combination with docetaxel, in 2022 [170]. Rezvilutamide has been approved in China since 2022 for metastatic HSPC with a high tumor burden [171]. Numerous clinical studies are ongoing for these compounds to broaden their indication space, either as single agents or in combination with other drugs.

Currently, there are also additional AR antagonists in clinical trials. Proxalutamide is in several phase 2 trials for the treatment of mCRPC [172].TRC-253 is an AR antagonist that has completed a phase 1/2a clinical study for the treatment of mCRPC [173].

Chemical structures of AR antagonists are shown in Figure 2.

### 5.2. Emerging Strategies for Inhibition of AR Signaling

In addition to classical small molecule inhibitors, new modalities that address AR signaling have emerged in recent years. Proteolysis targeting chimeras (PROTACs) are bifunctional molecules that recruit a cellular E3 ligase to drive ubiquitylation and the degradation of a target protein [174]. Several PROTACs targeting the AR have been reported, the most advanced of which being bavdegalutamide (ARV-110), which is currently in phase 2 clinical trials (Figure 3) [175]. It is hoped that the PROTAC-mediated degradation of the AR will yield clinical benefits beyond that which can be achieved with traditional small molecule antagonists, thus overcoming or delaying resistance mechanisms commonly seen with other AR-targeting drugs [176].

Other research groups have sought to develop new AR small molecules that can bind to the N-terminal domain (NTD) rather than to the ligand-binding domain (LBD) [177]. In contrast to the LBD, the NTD is considered far more difficult to drug, owing to an intrinsically disordered structure and lack of a suitable binding pocket. However, NTD binders could potentially retain activity in AR splice variants in which the LBD has been truncated. AR-V7 and other AR splice variants have been linked to worsened prognosis and a lack of druggable LBD, however, it may be druggable using NTD binder drugs [177]. Several compounds aiming to inhibit AR function through binding to the NTD have been described [178,179]. The first of these agents was ralaniten acetate (EPI-506, Figure 4), which has however been superseded by a next-generation compound named EPI-7386 which entered a phase 1/2 clinical trial in late 2021 (structure not disclosed). An NTD binder that drives AR degradation, named EPI-8207, has also been described, but its chemical structure is not yet disclosed [180]. Another recently described AR NTD binder and degrader is ONCT-534, formerly named GTx-534 (structure not disclosed) [181]. It showed strong efficacy in a patient-derived prostate cancer model expressing AR splice variants. While potentially promising, it remains to be seen how these agents will perform in advanced clinical studies.

## 6. PI3K/AKT/mTOR Inhibitors

### 6.1. Clinically Advanced PI3K/AKT/mTOR Inhibitors

#### 6.1.1. FDA-Approved and Clinically Advanced PI3K Inhibitors and Dual PI3K/mTOR Inhibitors

Small molecule inhibitors for PI3K can either inhibit multiple isoforms (pan-PI3K inhibitors) or be isoform-specific. Additionally, such drugs may be dual inhibitors that can also target mTOR. In this section, we focus on FDA-approved and clinically advanced PI3K inhibitors which have moved to or beyond phase 2 clinical trials (Figure 5 and Figure 6).

Idelalisib is specific for the δ-isoform and was the first PI3K inhibitor to be approved by the FDA in 2014 for the treatment of chronic lymphocytic lymphoma (CLL) [182,183].

Copanlisib is a pan-class I PI3K inhibitor, which has received orphan drug and fast-track designation by the FDA [184]. It was approved for the treatment of relapsed/refractory follicular lymphoma (FL) in 2017. A phase 1b/2 study combining copanlisib with the PARP inhibitor rucaparib is currently ongoing in mCRPC [185]. In addition, copanlisib has been included in a clinical phase 2 study enrolling patients with activating PI3Kα mutations and has shown activity based on an objective response rate [186].

Alpelisib is an α-isoform-specific PI3K inhibitor that has received orphan drug designation by the FDA. It was approved in combination with the selective ER degrader fulvestrant for the treatment of hormone receptor (HR)-positive, human epidermal growth factor receptor 2 (HER2)-negative and PI3Kα-mutated breast cancer at an advanced or metastatic stage in 2019 [187].

Umbralisib is a δ-isoform-specific PI3K inhibitor that was approved for the treatment of marginal zone lymphoma (MZL) and FL in 2021 [188,189]. The compound was recently withdrawn from the market due to the possibility of an increased risk of death [190].

Duvelisib targets the γ and δ isoforms of PI3K [191]. It was approved in 2018 for the treatment of relapsed or refractory CLL and small lymphocytic lymphoma (SLL).

In addition to FDA-approved drugs, there are also several PI3K inhibitors that have progressed to advanced clinical trials. Dactolisib is a pan-PI3K inhibitor that also inhibits mTOR [192] and was the first PI3K inhibitor to enter clinical trials. It was investigated in phase 3 clinical trials for clinically symptomatic respiratory illness and respiratory tract infections. Additionally, there were several phase 2 clinical trials for the treatment of various cancers, including metastatic breast cancer (mBC) [193,194]

Gedatolisib is a dual-form PI3K inhibitor that addresses the different PI3K isoforms while also targeting mTOR [195]. It showed encouraging results when combined with palbociclib, an inhibitor of the cyclin-dependent kinases CDK4 and CDK6, for the treatment of estrogen receptor (ER)-positive, HER2-negative mBC and is currently being investigated in phase 3 clinical trial [196]. Gedatolisib is also undergoing several phase 1/2 clinical trials as a treatment option for different solid tumors, including triple-negative breast cancer (TNBC) [197].

Paxalisib is a dual form pan-PI3K inhibitor that furthermore targets mTOR [198,199]. Paxalisib received orphan drug designation by the FDA in 2022 for the treatment of atypical rhabdoid or teratoid tumors in rare and aggressive childhood brain cancer. The therapeutic lead indication of paxalisib is glioblastoma, and it is currently being investigated in phase 2 and 3 clinical trials [200].

Buparlisib was developed as a pan-PI3K inhibitor and received fast-track designation by the FDA in combination with the tubulin-targeting drug paclitaxel for the treatment of recurrent or metastatic head and neck squamous cell carcinoma (HNSCC) [201,202]. It was evaluated in clinical trials for different additional indications, including breast and prostate cancer [203,204].

Samotolisib is a dual PI3K/mTOR inhibitor that was originally assessed in patients with diverse solid tumors [205]. It is currently being evaluated in mCRPC in combination with enzalutamide [206].

AZD8186 is a selective PI3K β/δ inhibitor that is being clinically tested in advanced solid tumors, including prostate and breast cancer [133].

Leniolisib is a δ isoform-specific PI3K inhibitor [207]. It has received orphan drug designation and priority review by the FDA for the treatment of activated phosphoinositide 3-kinase delta syndrome (APDS) a rare genetic immunodeficiency disease [208]. This treatment option is currently being evaluated in phase 3 clinical trials.

Zandelisib is another PI3K inhibitor with selectivity for the δ isoform [209]. It was initially evaluated in patients with B-cell malignancy, but development outside Japan was recently discontinued.

Eganelisib targets the γ isoform of PI3K [210]. The FDA has granted fast-track designation for the treatment of TNBC in combination with the PD-1-targeting monoclonal antibody nivolumab [211].

GSK2636771 is a phase 2 clinical candidate that targets the β isoform of PI3K [212]. It is being evaluated as a treatment option for different indications, including prostate cancer [213].

Inavolisib (GDC-077) is an α-isoform-targeting PI3K inhibitor currently in phase 3 trials for breast cancer. It not only inhibits PI3Kα but also induces the degradation of its target protein [214]. At the time of writing, no prostate cancer trials have yet been reported for this compound.

#### 6.1.2. AKT Inhibitors

In contrast to PI3K inhibitors, no small molecule inhibitors for AKT have been approved by the FDA. AKT inhibitors can either be allosteric or ATP-competitive. In this section, we have focused on clinically advanced AKT inhibitors, which are in phase 2 or beyond (Figure 7).

MK-2206 is an allosteric AKT inhibitor that targets the isoforms 1, 2, and 3. It was evaluated in clinical trials for the treatment of different tumor types, including breast, endometrial and uterine cancer, but the clinical benefit was limited [215,216,217,218,219].

The allosteric pan-AKT inhibitor TAS-117 is a clinical candidate for advanced solid tumors with germline *PTEN*-inactivating mutations. It is being investigated in phase 2 clinical trials for advanced or metastatic solid tumors [220,221].

Afuresertib is an ATP-competitive AKT inhibitor that targets the isoforms 1, 2, and 3 [222]. It is in phase 2 clinical trials for the treatment of various cancers, including non-small cell lung cancer, cervical cancer, TNBC, and prostate cancer [223,224].

Capivasertib is also an ATP-competitive pan-AKT inhibitor that is currently being evaluated in several phase 3 clinical trials for the treatment of solid tumors, including prostate [225,226] and breast [227] cancer. In addition, phase 2 studies are ongoing for ovarian and endometrial cancer [228], as well as for bladder cancer, meningioma, and non-Hodgkin lymphoma.

Another ATP-competitive pan-AKT inhibitor which is currently in the phase 3 clinical-stage is ipatasertib [229]. Similarly to capivasertib, it is being evaluated in clinical trials for the treatment of prostate [230,231,232] and breast [233,234,235] cancer, as well as in other tumor types.

#### 6.1.3. mTOR Inhibitors

The first mTOR inhibitors were originally approved for the prevention of graft rejection [236]. They are allosteric inhibitors derived from the natural macrolide rapamycin (sirolimus) and include temsirolimus and everolimus (Figure 8). Later, small molecules acting as ATP-competitive inhibitors and dual-binding site inhibitors have been described. Currently, there are two mTOR inhibitors that have been approved for different cancer indications with, however, limited efficacy [237]. Therefore, the combined PI3K/mTOR blockade might be more effective (see above). Several clinical studies combining mTOR inhibitors with AR signaling inhibitors are currently ongoing, but dose-limiting toxicity has been observed [70]. Detailed reviews on mTOR inhibitors have recently been published [133,238,239].

### 6.2. Emerging Strategies for PI3K/AKT/mTOR Inhibition

In addition to the identification of classical small molecule inhibitors, there have been recent advances in the application of new therapeutic modalities to address the PI3K/AKT/mTOR signaling axis. Among the most advanced of these are allosteric inhibitors of PI3Kα, which are claimed to target H1047x oncogenic mutants while sparing wild-type PI3Kα. It is hoped that mutant selectivity will improve upon the safety profile and therapeutic index of wild-type PI3Kα inhibitors. To our knowledge, three mutant-selective allosteric small molecules are currently under development. LOXO-783 is claimed to target the H1047R mutant and shows antiproliferative activity in different breast cancer models, leading to the initiation of phase 1 clinical study [240]. RLY-2608 is purported to be pan-PI3Kα mutant selective. It inhibits AKT phosphorylation and the viability of cell lines originating from breast cancer and other tumors. This compound has just started a phase 1 clinical trial [241]. STX-478 is also a pan-PI3Kα mutant-selective compound and shows potent activity in several tumor types [242]. Although most frequent in breast cancer, PI3Kα mutations are also found in mCRPC, where they can be involved in worse disease progression, which suggests that allosteric PI3Kα mutant-selective inhibitors may find future applications for the treatment of prostate cancer [96]. While the structures of the mentioned clinical candidate molecules have not been disclosed, representative structures taken from the patent literature are presented in Figure 9.

Targeted protein degradation is another emerging therapeutic modality that has been applied to address the PI3K/AKT/mTOR pathway. A PI3K/mTOR-targeting PROTAC (Compound HL-8, Figure 10) showing strong degradation of PI3K kinase at a concentration of 10 µM has been reported [243]. At the same concentration, this compound also showed significant phosphorylation inhibition of the downstream marker AKT, and this was coupled with anti-proliferative effects in HeLa cells.

AKT kinase has also been targeted using the PROTAC modality. Two research teams independently reported the identification of AKT PROTACs, one named INY-03-041 and the other one MS98, both of which employ the same AKT-targeting moiety but which differ in the E3 ligase binding component (Figure 11) [244,245]. Both compounds are effective degraders of AKT. They inhibit downstream signaling, leading to the prolonged inhibition of the AKT pathway, and have antiproliferative effects in diverse tumor cell lines, including prostate cancer models.

At the time of writing, there are no clinical trials reporting the use of PROTAC degraders of the PI3K/AKT/mTOR pathway. However, the clinical PI3Kα inhibitor inavolisib (GDC-077) has been purported to drive the degradation of mutant PI3Kα, preventing negative feedback loops and driving prolonged pathway inhibition in PI3Kα mutant preclinical models [214,246]. However, the mechanism of degradation is not yet clear. Inavolisib is under clinical development for breast cancer, with no published applications related to AR signaling or for the indication of prostate cancer. Nonetheless, given the potential benefits and differentiation of protein degradation versus traditional small molecule inhibition, the future may bring further PI3K/AKT/mTOR degraders into clinical development.

## 7. Recent Clinical Studies Combining Inhibitors of the PI3K/AKT/mTOR and AR Pathways for Prostate Cancer Treatment

Different compounds inhibiting the PI3K/AKT/mTOR pathway have been evaluated or are currently undergoing clinical studies in prostate cancer patients, mainly at the mCRPC stage [21,71,133,134,247,248,249]. However, reciprocal feedback loops leading to the activation of resistance mechanisms and dose-limiting side effects have so far limited the impact of these treatments. Combination therapies with AR inhibitors are now being evaluated in late-stage clinical studies (Table 1). Examples of advanced studies with results include the combination of the PI3K inhibitor samotolisib with enzalutamide, which causes an improved progression-free survival in mCRPC patients progressing on abiraterone [206]. Another example is the phase 3 study combining the AKT inhibitor ipatasertib with abiraterone acetate, which has shown a significant positive impact on progression-free survival in mCRPC patients with *PTEN* loss [230].

## 8. Conclusions and Perspectives

Both the AR and the PI3K/AKT/mTOR pathways are essential players in prostate cancer, and reciprocal feedback has been outlined in numerous studies. AR signaling inhibitors are established therapies for different stages of the disease, and combination treatments with other targeted agents are currently being clinically tested. Five PI3K inhibitors with different isoform selectivity profiles are now approved, mainly for the treatment of different leukemia and lymphoma forms and for breast cancer, whereas two mTOR inhibitors are marketed mainly for renal cell carcinoma and breast cancer [71,72]. Several other compounds, including dual PI3K/mTOR inhibitors and AKT inhibitors, are currently in clinical studies and are being evaluated for monotherapy or in combination treatments [21,71,72,90,133,247,248].

Concerning prostate cancer, the most advanced study reports that the AKT inhibitor ipatasertib, when given in combination with abiraterone acetate, leads to a significant improvement in the radiographic progression-free survival of mCRPC patients with *PTEN* loss [230]. Several other PI3K/AKT/mTOR pathway inhibitors have been evaluated in clinical trials, but in many instances, the occurrence of adverse events was dose-limiting, and no clear benefit could be observed [71,248]. More targeted approaches with compounds addressing the tumor-specific p110α mutations are currently in the early phases of clinical development and may constitute a more promising therapeutic strategy [250]. Additionally, compounds leading to the cellular degradation of PI3K or AKT when using the PROTAC approach and which may have superior efficacy have been described [143,243,251,252].

In view of the dose-limiting effects observed in the clinic with most PI3K/AKT/mTOR pathway inhibitors, it will be essential to identify stratification biomarkers to determine which patient groups are most likely to benefit from these treatments. This may prove challenging in prostate cancer due to the high inter- and intra-tumor heterogeneity, which makes the investigation of the precise status difficult. Recent advances include the use of artificial intelligence-based approaches for the automated detection of *PTEN* loss in prostate cancer samples stained by immunohistochemistry [253]. Additionally, liquid biopsies are now taken to purify circulating tumor DNA for subsequent analysis by deep sequencing [254]. This should soon help to decide on improved treatment strategies for the benefit of prostate cancer patients.

## Figures and Tables

**Figure 1 ijms-24-02289-f001:**
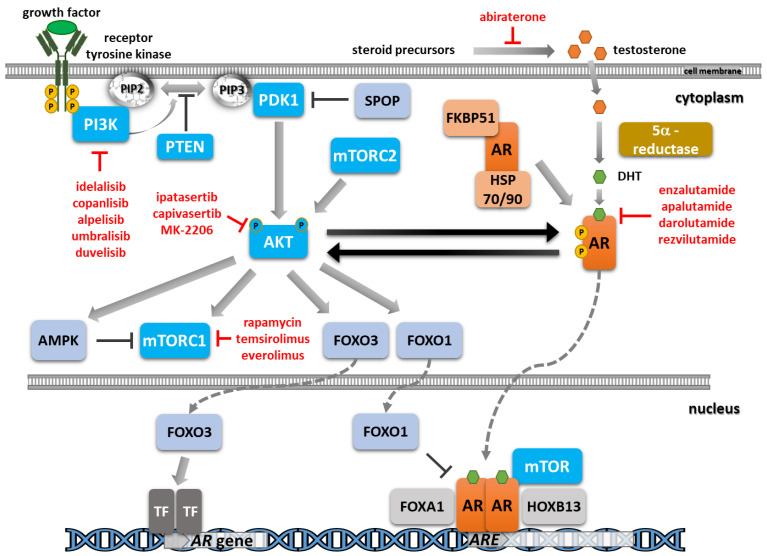
Schematic overview of the AR and PI3K/AKT/mTOR signaling pathways and their crosstalk. Selected approved and advanced inhibitors addressing these pathways are indicated.

**Figure 2 ijms-24-02289-f002:**
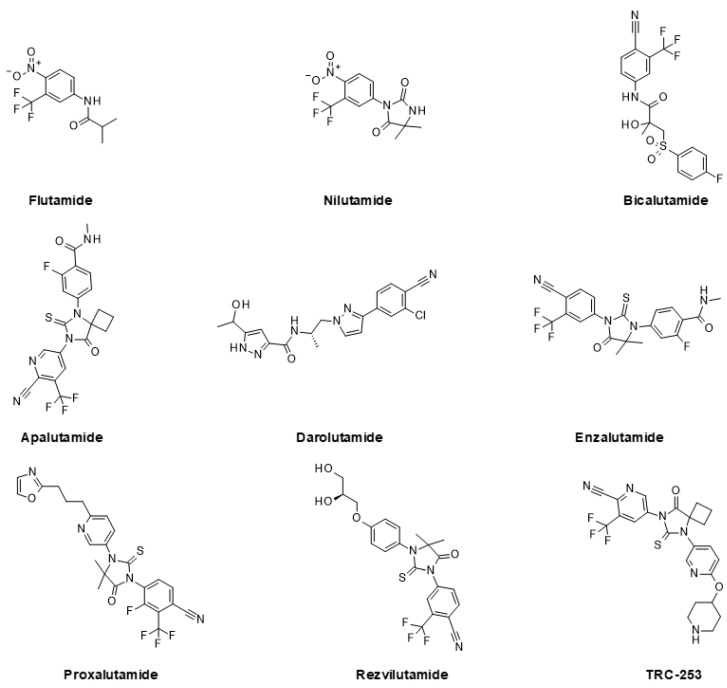
AR inhibitors launched or in late-stage clinical trials.

**Figure 3 ijms-24-02289-f003:**
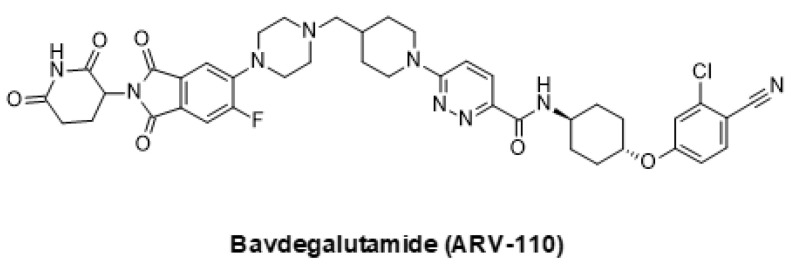
The AR-targeted PROTAC bavdegalutamide (ARV-110).

**Figure 4 ijms-24-02289-f004:**
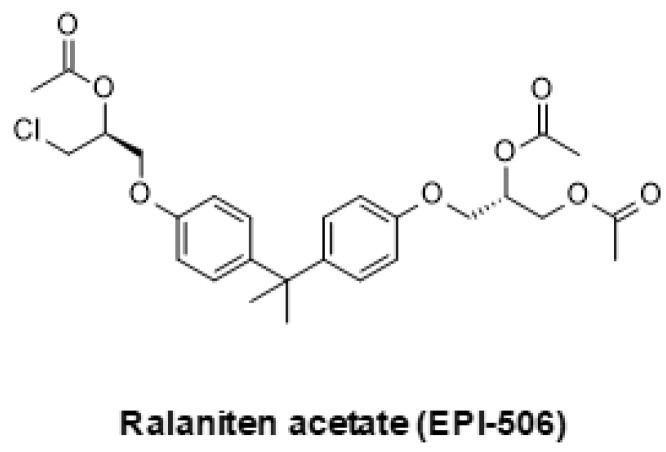
The AR-V7 inhibitor EPI-506.

**Figure 5 ijms-24-02289-f005:**
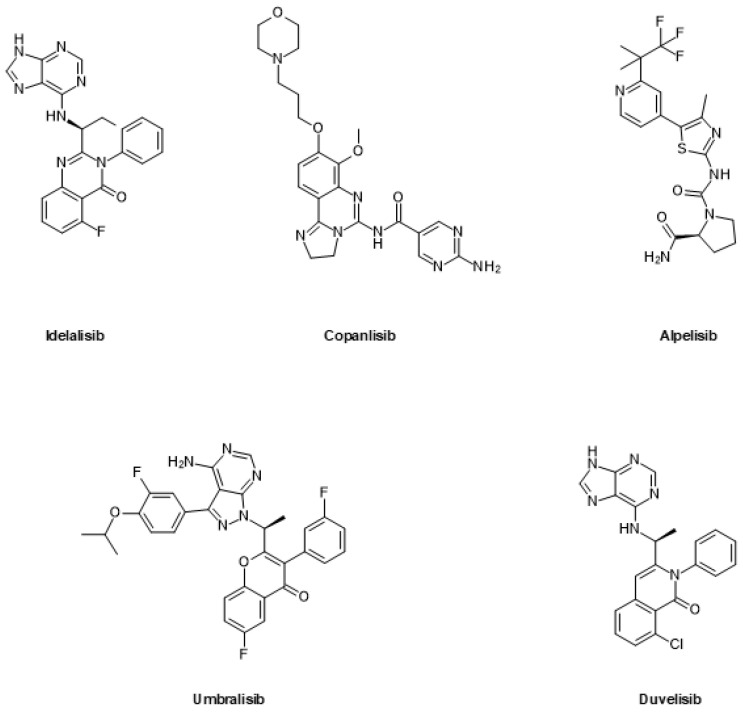
FDA-approved inhibitors of PI3K.

**Figure 6 ijms-24-02289-f006:**
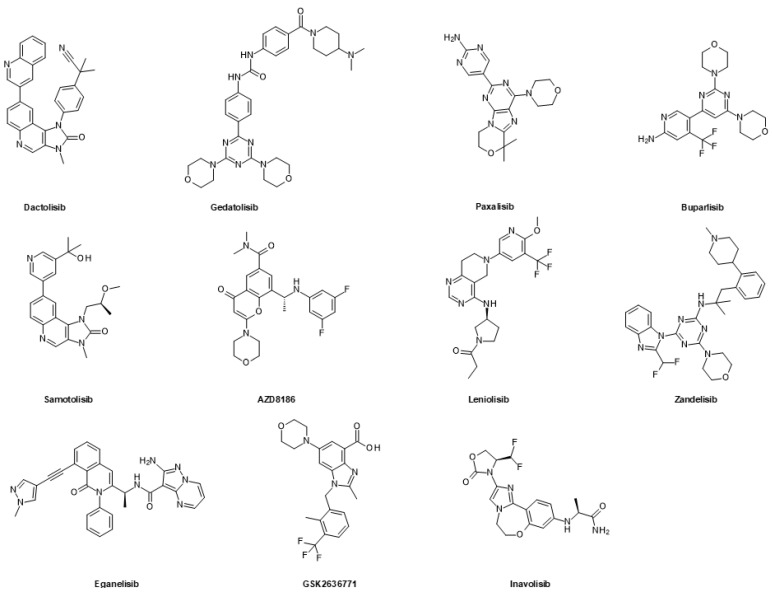
Inhibitors of PI3K in advanced clinical trials.

**Figure 7 ijms-24-02289-f007:**
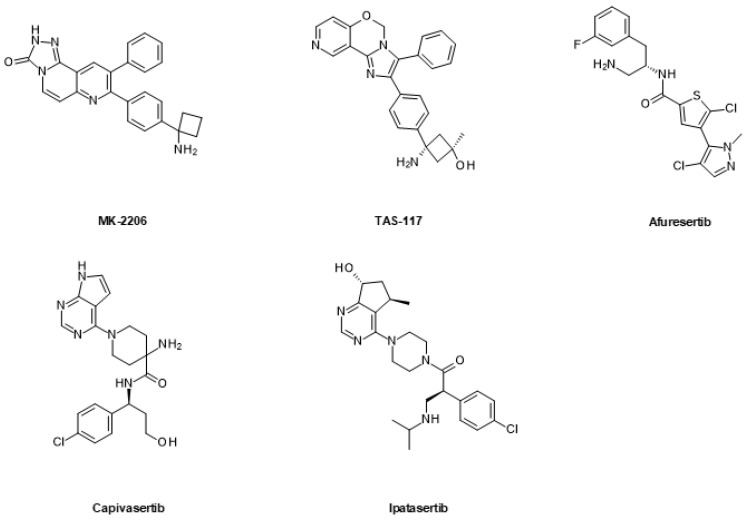
Inhibitors of AKT in advanced clinical trials.

**Figure 8 ijms-24-02289-f008:**
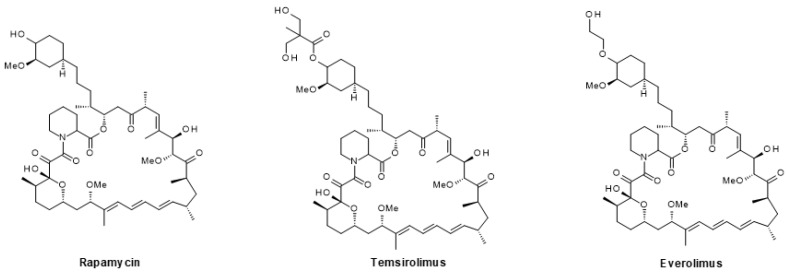
Macrocyclic allosteric mTOR inhibitors.

**Figure 9 ijms-24-02289-f009:**
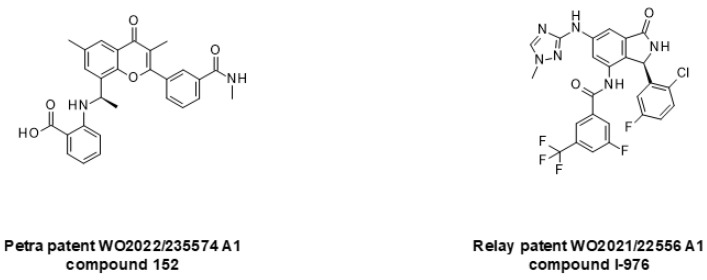
Mutant-selective allosteric inhibitors of PI3Kα.

**Figure 10 ijms-24-02289-f010:**
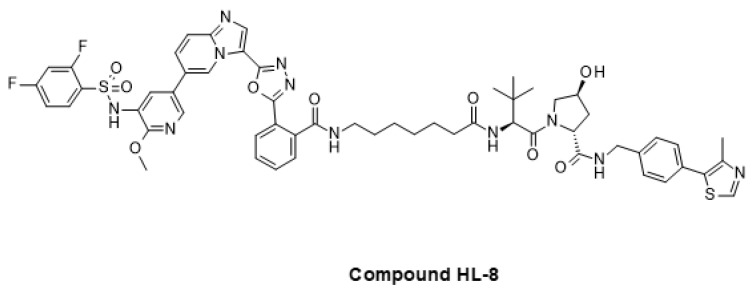
Compound HL-8: a recently reported PROTAC degrader of PI3K.

**Figure 11 ijms-24-02289-f011:**
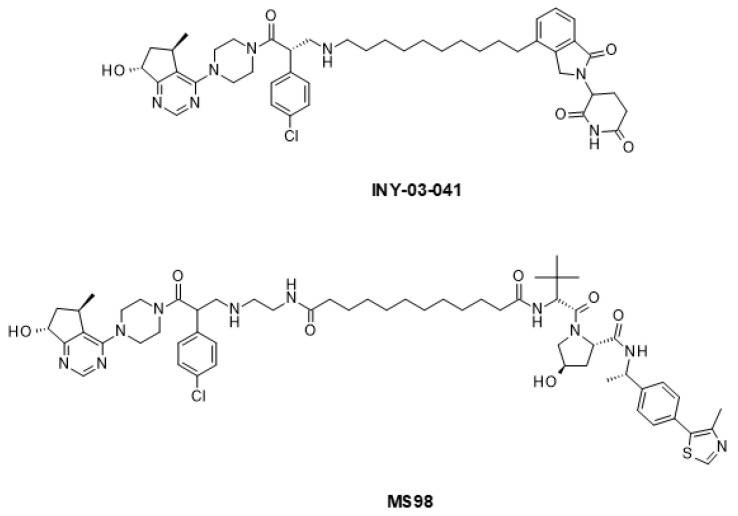
PROTAC degraders of AKT.

**Table 1 ijms-24-02289-t001:** Selection of prostate cancer clinical trials combining PI3K/AKT/mTOR and AR pathway inhibitors.

Drug Combination	Targets	Indication	Status	Phase	Identifier
Samotolisib + enzalutamide	PI3K-mTOR/AR	mCRPC	Completed	2	NCT02407054
Buparlisib + enzalutamide	PI3K/AR	mCRPC	Terminated	2	NCT01385293
Apitolisib + abiraterone acetate	PI3K-mTOR/CYP17A	CRPC	Completed	1b/2	NCT01485861
GSK2636771 + enzalutamide	PI3K/AR	mCRPC	Completed	1	NCT02215096
AZD8186 + abiraterone acetate	PI3K/CYP17A	Advanced CRPC	Completed	1	NCT01884285
Ipatasertib + abiraterone acetate	AKT/CYP17A	mCRPC	Active, not recruiting	3	NCT03072238
Capivasertib + abiraterone acetate	AKT/CYP17A	mHSPC with *PTEN* deficiency	Recruiting	3	NCT04493853
MK-2206 + bicalutamide	AKT/AR	Recurrent prostate cancer	Active, not recruiting	2	NCT01251861
Capivasertib + abiraterone acetate	AKT/CYP17A	High-risk localized prostate cancer with *PTEN* deficiency	Not yet recruiting	2	NCT05593497
Capivasertib + enzalutamide	AKT/AR	mCRPC	Unknown	2	NCT02525068
Ipatasertib + abiraterone acetate	AKT/CYP17A	CRPC	Completed	1b/2	NCT01485861
Ipatasertib + darolutamide	AKT/AR	Localized CRPC with *PTEN* deficiency	Terminated	1/2	NCT04737109
Everolimus + bicalutamide	mTOR/AR	Recurrent or metastatic CRPC	Completed	2	NCT00814788
Everolimus + enzalutamide	mTOR/AR	mCRPC	Completed	1	NCT02125084
Everolimus + apalutamide	mTOR/AR	mCRPC	Completed	1	NCT02106507

## Data Availability

Not applicable.

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
