# Peer review of "Addressing the Reciprocal Crosstalk between the AR and the PI3K/AKT/mTOR Signaling Pathways for Prostate Cancer Treatment"

_ijms, 2023, doi:10.3390/ijms24032289_

Round 1

Reviewer 1 Report

Comments to authors

This study evaluated the role of reciprocal crosstalk between AR and PI3K/AKT/mTOR signaling pathways for prostate cancer treatment. The authors have addressed all the issues more or less adequately, they also included the most recent papers found in the literature and presented a very thorough review. The manuscript is well-organized and written and I don’t have any remark to indicate. In my opinion this paper should be accepted without any revisions.

Author Response

Many thanks for the positive feedback.

Reviewer 2 Report

In this review Raith et al., discussed the interplay between PI3K/AKT/mTOR pathway and AR signaling in the evolution of prostate cancer and its treatment approaches. 

Suggestions to further improve the manuscript:

1. To help readers to connect/understand well - please include a figure depicting the cross talk between PI3K/AKT/mTOR pathway and AR signaling:  if possible, with targeting agents in development for the treatment of resistant prostate cancer.

2. Similar to the section 2 in the manuscript: where authors discussed about the PI3K/AKT/mTOR signaling in prostate cancer - It would be logical to include a dedicated section to AR signaling in prostate cancer. Where authors can explain (i) AR pathway overview (ii) AR pathway alterations in prostate cancer.

3. Curious to know why authors not discussed about treatment resistant Neuro Endocrine Prostate Cancer model - Where AR signaling is suppressed and AKT signaling is activated.

4. Please comment on the following manuscripts: 

4a. MAPK4 promotes prostate cancer by concerted activation of androgen receptor and AKT PMID: 33586682- Where authors showed that MAPK4 promotes prostate cancer growth by concerted activation of AR and AKT signaling pathways.

4b. Dual inhibition of AKT-mTOR and AR signaling by targeting HDAC3 in PTEN - or SPOP-mutated prostate cancer PMID: 29523594 - where authors identified HDAC3 as a common upstream activator of AKT and AR signaling pathways.

Author Response

Many thanks for your positive feedback and constructive comments. Here how we addressed your suggestions:

1) A figure depicting the crosstalk between PI3K/AKT/mTOR pathway and AR signaling, including the most advanced targeting agents has been added as a new figure 1.

2. A dedicated section on AR signaling in prostate cancer, including the pathway overview, alterations seen in prostate cancer, and also a paragraph on NEPC has been added.

3. NEPC is now addressed in paragraphs 1 and 2.3, and at the end of paragraph 3.3.

4. Please comment on (1) MAPK4 promotes prostate cancer by concerted activation of androgen receptor and AKT PMID: 33586682 and (2) Dual inhibition of AKT-mTOR and AR signaling by targeting HDAC3 in PTEN - or SPOP-mutated prostate cancer PMID: 29523594: This was added to paragraph 4.2

In addition, we mention in paragraph 5 that the antiandrogen rezvilutamide is approved in China. We also added a new paragraph 6.1.3 and a new figure 8 on mTOR inhibitors, as they are also being clinically tested in prostate cancer, as already mentioned in table 1.

Please note that all these changes are highlighted in red in the text.